# Long-Term Exposure to Decabromodiphenyl Ether Promotes the Proliferation and Tumourigenesis of Papillary Thyroid Carcinoma by Inhibiting TRß

**DOI:** 10.3390/cancers14112772

**Published:** 2022-06-02

**Authors:** Xinpei Wang, Xiujie Cui, Qian Zhao, Feifei Sun, Ru Zhao, Tingting Feng, Shaofeng Sui, Bo Han, Zhiyan Liu

**Affiliations:** 1The Key Laboratory of Experimental Teratology, Ministry of Education and Department of Pathology, School of Basic Medical Sciences, Shandong University, Jinan 250012, China; wang-xinpei@mail.sdu.edu.cn (X.W.); cuixiujie0826@163.com (X.C.); 201820521@mail.sdu.edu.cn (F.S.); 201715012@mail.sdu.edu.cn (R.Z.); 202190000038@sdu.edu.cn (T.F.); boh@sdu.edu.cn (B.H.); 2Department of Pathology, Shanghai Jiao Tong University Affiliated Sixth People’s Hospital, 600# Yishan Rd, Shanghai 200233, China; zhaoqian9168@163.com; 3Department of Pathology, The Second Hospital, Shandong University, Jinan 250033, China; 4Department of Environmental Health, Division of Health Risk Factors Monitoring and Control, Shanghai Municipal Center for Disease Control and Prevention, State Environmental Protection Key Laboratory of Environmental Health Impact Assessment of Emerging Contaminants, 1380 West Zhongshan Road, Shanghai 200336, China; suishaofeng@scdc.sh.cn; 5Department of Pathology, Qilu Hospital, Shandong University, Jinan 250012, China

**Keywords:** papillary thyroid carcinoma, BDE209, chronic toxicity, tumourigenesis, TRß

## Abstract

**Simple Summary:**

PBDEs have been reported to have endocrine-disrupting and tumour-promoting activity; however, the effects of BDE209 (the highest brominated PBDEs) on the thyroid and the underlying mechanisms are unclear. In this study, we found that long-term exposure to BDE209 could cause chronic toxicity and potential tumourigenesis by inhibiting the expression and function of TRß, which induces the proliferation of thyroid tissue and the oncogenesis of thyroid carcinoma. These findings emphasize the damaging effects that exposure to BDE209 has on human thyroid and papillary thyroid carcinoma.

**Abstract:**

Polybrominated diphenyl ethers (PBDEs) have been reported to possess endocrine-disrupting and tumour-promoting activity. However, the effects of long-term exposure to decabromodiphenyl ether (BDE209) on thyroid tumourigenesis of papillary thyroid carcinoma (PTC) and the underlying mechanisms remain poorly defined. In this study, functional assays in vitro and mouse models in vivo were used to evaluate the toxic effects of long-term exposure to environmental concentrations of BDE209 on the pathogenesis and progression of PTC. MTS, EdU and colony-forming assays confirmed the chronic toxicity of BDE209 on the proliferation of human normal follicular epithelial cell line (Nthy-ori 3-1) and PTC-derived cell lines (TPC-1 and BCPAP). Wound and Transwell assays showed that BDE209 exacerbated the aggressiveness of PTC cells. BDE209 significantly promoted cell proliferation during the S and G2/M phases of the cell cycle. Mechanistically, BDE209 altered the thyroid system by acting as a competitive inhibitor of thyroid receptor beta (TRß) expression and function, which was further proven by public databases and RNA-seq bioinformation analysis. Taken together, these results demonstrated that BDE209 has chronic toxicity and potential tumourigenic effects on the thyroid by inhibiting TRß.

## 1. Introduction

Thyroid carcinoma is one of the most common endocrine malignancies, with an estimated 600,000 new cases annually worldwide, and papillary thyroid carcinoma (PTC) comprises approximately 90% of all cases [1,2]. High-grade PTC presents a more aggressive clinical course than poorly differentiated thyroid carcinoma [3,4], and approximately 30% of these tumours eventually progress to be locally advanced, metastatic, or refractory to radioactive iodine [5,6,7]. PTCs are generally driven by only a few somatic mutations or mutually exclusive fusions involving genes regulating the mitogen-activated protein kinase (MAPK) signalling cascade [4,8,9]. However, the pathogenesis of PTC has not been completely elucidated thus far. Exposure to environmental contaminants has been reported to contribute to different kinds of diseases and cancers [10,11,12].

Polybrominated diphenyl ethers (PBDEs) are persistent organic pollutants (POP) that possess neurotoxicity, hepatotoxicity, immunotoxicity, reproductive toxicity, endocrine-disrupting activity and carcinogenicity [13,14,15]. Decabromodiphenyl ether (BDE209) is the most commonly used PBDEs and it has been widely detected in multiple environmental media and biological samples ranging from plankton to mammals [16,17,18]. As the most important PBDEs in serum and plasma, BDE209 has been demonstrated to be associated with thyroid dysfunction and thyroid carcinoma risk [18,19]. However, the exact effects and underlying mechanisms of this remain unknown. In this study, we investigated the effect and the potential molecular mechanisms of BDE209 on the development and progression of thyroid carcinoma.

## 2. Materials and Methods

### 2.1. Cell Culture and Reagents

The human normal follicular epithelial cell line Nthy-ori 3-1 was purchased from JENNIO Biological Company (Guangzhou, China). PTC cell lines TPC-1, BCPAP (with the *BRAF*^V600E^ mutation), and Liothyronine (HY-A0070A) were kindly provided by our collaborators [20,21]. HEK293T cells (CRL-3216) were obtained from the American Type Culture Collection (ATCC) (Rockville, MD, USA). BDE209 was dissolved in dimethyl sulfoxide (DMSO) (D2650 Sigma-Aldrich, St Louis, MO, USA). Culture medium without BDE209 and DMSO was used as a negative control.

### 2.2. Cell Proliferation Assays

Cells were treated with different concentrations of BDE209 for 24 h, 48 h, 72 h and 96 h and cell proliferation was measured as reported [22]. The proliferation and viability of different BDE209-treated groups were compared to nontreated control groups.

For the colony-forming assay, the cells were cultured with media containing 1 μM and 2.5 μM BDE209 for 10 days. They were fixed and stained violet before being photographed [22].

Cell-Light™ EdU DNA Cell Proliferation (EdU) assays (Ribobio Biotechnology, Guangzhou, China) were used to measure the cell proliferation. The stained cells were examined with a fluorescence microscope and photographed with a camera. ImageJ was used to count the number of all cells and proliferating cells.

### 2.3. Cell Cycle and Cell Apoptosis Determination by Flow Cytometry

Nthy-ori 3-1, TPC-1 and BCPAP cells were treated with 1 μM and 2.5 μM BDE209 for 27 days, synchronized in G0 by serum starvation for 24 h and then treated with different doses of BDE209 for 48 h. Cell cycle analysis was performed using propidium iodide staining (BestBio, Shanghai, China). The cells percentage in each cell cycle stage was calculated by Modfit LT 5 software.

### 2.4. Migration and Invasion Assays

The cells were incubated in culture medium with 2% FBS and corresponding concentration of BDE209. Differences in cell migration were quantified by comparing the wound healing areas after 24 h in at least five fields using ImageJ (NIH, Bethesda, MD, USA).

Transwell invasion assays were performed in 24-well plates with 8 μm pore size chamber inserts (Corning, NY, USA). Cells on the lower surface were stained and imaged (five independent fields per well) under a light microscope (ZEISS, Oberkochen, Germany).

### 2.5. Quantitative Real-Time PCR (qRT–PCR)

A ReverTra Ace qPCR RT kit (Toyobo, Japan) and SYBR Green PCR kit (Roche, UK) were used to examine the mRNA levels. GAPDH was used as the endogenous control for mRNA. The sequences of the primers are shown in Appendix A.

### 2.6. Subcellular Protein Extraction and Western Blot Analysis

To determine the subcellular localization of TRß, a nuclear/cytoplasmic fractionation kit (Beyotime, Haimen, China) was used. After being cultured with different treatments, the cytoplasmic and nuclear protein fractions of the cells were collected. Anti-GAPDH (1:1000, cat no. ab181602; Abcam) or anti-Lamin A/C (1:1000, cat no. 2032; Cell Signaling Technology) was used as control, respectively. Western blotting was performed as previously described [22,23]. Anti-TRß (1:1000, cat no. PA2163; Abmart) was used as the primary antibody.

### 2.7. Immunofluorescence and Confocal Imaging

Differently treated Nthy-ori 3-1 cells were incubated with primary antibodies against TRß (1:500, cat no. PA2163; Abmart) and Alexa Fluor 488-conjugated secondary antibodies. The expressions of TRß were analysed and imaged using a laser confocal microscope (LSM880, ZEISS, Oberkochen, BW, Germany).

### 2.8. Transient Transfection and Dual-Luciferase Reporter Assay

Luciferase reporter vectors containing the TRE (vector: pGL3) were constructed by Biosune Biotech (Shanghai, China). pTRE-TA promoter sequences are shown in Appendix A. Constructs of the pTRE-TA firefly luciferase reporter were co-transfected with pRL-TK Renilla luciferase reporter vector expression plasmids into 293T cells. Sixteen to 24 h after transfection, the wells were incubated with the indicated concentrations of ligand and/or BDE209 for 48 h. Bioactive T3 was used for thyroid hormone (TH) treatment.

### 2.9. RNA Sequencing and Bioinformatics Analysis

RNA sequencing analyses (BGI, Shenzhen, China) were performed on the control and BDE209-treated Nthy-ori 3-1 cells for 27 days. The differentially expressed genes were analysed using Gene Set Enrichment Analysis (GSEA) (http://software.broadinstitute.org/gsea/index.jsp, accessed on 27 March 2022). The relationship between *THRB* expression and thyroid tumours were analysed using datasets downloaded from the GEO database (http://www.ncbi.nlm.nih.gov/geo, accessed on 27 March 2022).

### 2.10. Tumour Xenografts

To investigate the role of BDE209 in thyroid and PTC cells, a total of 2.0 × 10^6^ BCPAP cells were mixed with Matrigel (1:1) and injected subcutaneously into the four-week-old female BALB/c mice (Weitonglihua Biotechnology, Beijing, China). They were randomized into two groups (*n* = 5/group) and treated as follows: 1, vehicle control (peanut oil, p.o.) and 2, BDE209 (2000 µg/kg/d, p.o.). Tumour tissues were harvested and weighed after 4 weeks of treatment. The tumour size was measured twice a week, and the tumour volume was calculated with the following formula: tumour volume = length × width^2^ × 0.5.

### 2.11. Histology and Immunohistochemistry

Dissected thyroid and tumour tissues were formalin-fixed paraffin-embedded. Anti-Ki67 (ready-to-use, ZA-0502; Zsbio) was used for immunohistochemistry (IHC). For assessment of the intensity, each field was graded semiquantitatively on a three-tier scale (0 = negative staining, 1 = weak staining, 2 = moderate staining, 3 = strong staining). Histology and IHC scores were evaluated blindly by two independent pathologists (Dr. X.C. and Dr. Z.L.).

### 2.12. Statistical Analysis

Statistical analysis was carried out using GraphPad Prism 9 (La Jolla, CA, USA). The two-tailed unpaired t-test was used to compare the difference between two groups. Tumour growth was analysed by ANOVA. *P* values were two-sided and *p* < 0.05 was considered statistically significant.

## 3. Results

### 3.1. Hormesis Effect of BDE209 on Nthy-ori 3-1 and PTC Cell Proliferation

As shown in Appendix A, environmentally relevant doses (0.5 to 5 μM) of BDE209 for 48 h significantly promoted Nthy-ori 3-1-cell proliferation, while higher concentrations of BDE209 inhibited cell proliferation in a concentration-dependent manner. The concentrations with proliferation-promoting effects (1 and 2.5 μM) were preliminarily identified, which were then used in the following experiments.

Both the normal human thyroid follicular epithelial cell line (Nthy-ori 3-1) and the PTC cell lines (TPC-1 and BCPAP) were dramatically affected by BDE209 treatment in a dose- (1 and 2.5 μM) (Figure 1A) and time-dependent (9 and 27 days) (Figure 1B) manner. We showed that prolonged BDE209 treatment (~1 month) increased the hormesis effect in Nthy-ori 3-1, TPC-1 and BCPAP cells. The long-term exposure cells were established by culturing Nthy-ori 3-1, TPC-1 and BCPAP cells in media containing increasing doses of BDE209 from 1 to 2.5 μM for 27 days. These cells were then maintained in culture media containing 1 and 2.5 μM BDE209 and were used in the following experiments. We observed that 1.0 and 2.5 μM BDE209 treatment resulted in a significant increase in the number and the size of colonies formed by Nthy-ori 3-1, TPC-1 and BCPAP cells (Figure 1C) compared with the control group and DMSO group. The colony numbers were calculated and are shown in the lower panel.

### 3.2. Cell Cycle Analysis of Nthy-ori 3-1 and PTC Cells after Exposure to BDE209

EdU assays showed that proliferation was significantly increased after treatment with 1 μM and 2.5 μM BDE209 for 27 days in Nthy-ori 3-1 and PTC cells compared with the control and DMSO groups (Figure 2A). To further elucidate the mechanism by which the environmentally relevant concentrations of BDE209 promote cell proliferation, we analysed the influence of BDE209 on cell cycle regulation. Nthy-ori 3-1 and PTC cells were treated with environmentally relevant concentrations of BDE209 for 27 days, and the percentage of cells in the G0/G1, S and G2/M stages were assayed by flow cytometry. As shown in Figure 2B, the BDE209-treated groups had an inordinately increased proportion of S- and G2/M-phase cells compared with the control and DMSO groups.

### 3.3. Hormesis Effect of BDE209 on Nthy-ori 3-1 and PTC Cell Invasion and Migration

We next investigated the migratory properties of BDE209-treated PTC cells using wound-healing assays (Figure 3A). The mobility of the carcinoma cells was significantly accelerated with environmentally relevant doses of BDE209 exposure as compared to the control and DMSO groups. Moreover, the Transwell system further confirmed the effects of BDE209 on TPC-1 and BCPAP cell migration and invasion. As shown in Figure 3B, BDE209 at 1 μM and 2.5 μM significantly increased the quantity of invading cells compared to the control. These results indicated that BDE209 promoted both the migration and invasion of PTC cancer cells.

### 3.4. BDE209 Promotes Tumourigenesis by Inhibiting TRß Expression and Function

The chemical structure of BDE209 closely resembles that of T3 and T4 and hydroxylated metabolites of BDE209 bind with high affinity to thyroid hormone receptors (TRs). We therefore wondered whether BDE209 treatment alters the thyroid system through hormone mimicry in human thyroid follicular epithelial cells. Thus, the expression of TRß in Nthy-ori 3-1 cells was detected after exposure to different concentrations of BDE209. As shown in Figure 4A,B and Appendix A, an environmentally relevant concentration of BDE209 decreased the mRNA and protein levels of TRß compared to the control and DMSO groups, and the levels of TRß decreased in a dose-dependent manner. To determine which subcellular localization of TRß is responsible for BDE209, we applied immunoblotting analysis. We observed that TRß could be expressed in both cytoplasmic and nuclear fractions of cells. However, the levels of both cytoplasmic and nuclear TRß were decreased when Nthy-ori 3-1 cells were treated with BDE209 for 27 days (Figure 4C and Appendix A). Consistent with our findings, in IF assays, TRß was expressed at low levels when the cells were treated with 1 μM and 2.5 μM BDE209 compared to the control groups (Figure 4D). TRs are DNA-binding transcription factors that recognize specific and highly conserved DNA sequences called TRE in the promoters of T3 target genes. We determined the effect of BDE209 on TRß-mediated transcription in 293T cells using a transient transfection-based dual-luciferase reporter gene assay. Our data showed that BDE209 suppressed TRβ-mediated transcription of TRE and significantly altered the transcription levels when T3 was present (Figure 4E and Appendix A). These results suggest that there may be competitive inhibition between BDE209 and T3 at multiple levels.

### 3.5. Inverse Association between Expression of TRß and Thyroid Cancer

To explore the clinical significance of the antagonistic activity of BDE209 on TRß in human thyroid tumours, we analysed the expression of *THRB* in multiple publicly accessible datasets. *THRB* expression was significantly decreased in thyroid tumour samples compared with that of the corresponding normal thyroid samples, as shown in different gene expression profile data. The data from GSE60542 also demonstrated that *THRB* was significantly downregulated in lymph node metastasis compared with the corresponding normal thyroid tissue. In the data from GSE53157, the downregulation of *THRB* mRNA was significantly associated with different types of thyroid cancers, including FTC, FVPTC, PDTC and PTC (Figure 5A). To further decipher the mechanism by which BDE209 promotes thyroid cancer, we performed RNA-seq analysis in Nthy-ori 3-1 cells. We compared the transcriptomes between the BDE209 group (treated with 2.5 μM BDE209 for 27 days) and the control group (treated with the same volume of DMSO for 27 days). GSEA revealed that TH response pathway genes (*p* = 0.0) and thyroid cancer-related genes (*p* = 0.001) were significantly enriched in the BDE209-treated group (Figure 5B). Taken together, BDE209 exposure is associated with the carcinogenic potential of thyroid cancer.

### 3.6. BDE209 Promotes the Proliferation of Thyroid Follicular Cells and the BCPAP Cells in BALB/c Female Mouse Xenograft Models

We examined the effect of BDE209 on the thyroid glands of BALB/c female mice. Mice were randomly assigned to be treated with peanut oil (control group) or peanut oil plus BDE209 (BDE209 group). H&E examination of the thyroid gland tissue samples revealed that the histological structure of the thyroid in the control group exhibited a generally normal morphology. Compared with the control group, the proliferation of the thyroid follicles was demonstrated morphologically in the high concentration 2000 µg/kg/d BDE209 group (Figure 6A). To examine the impact of BDE209 on tumour growth, we established BCPAP xenografts in BALB/c female mice. BDE209 induced the proliferation of the tumours as confirmed by tumour size and weight (Figure 6B–C) and the proliferation index of ki-67 (Figure 6D). However, the mouse body weights did not change in any of the groups. These results suggest that BDE209 may induce the proliferation of normal thyroid follicular cells or promote their oncogenicity and the growth of thyroid papillary carcinoma cells. The working model is shown in Figure 6E.

## 4. Discussion

BDE209 has been extensively added to various commercial products and can easily volatilize and leak into the surrounding environment [24]. It has been listed as a POP under the Stockholm Convention. Nevertheless, the production and utilization of BDE209 are continuing [25]. The concentration of BDE209 in serum was reported to be as high as 3100 ng/g lipid weight (approximately 3.2 μM) in residents in production areas, which correlates with thyroid dysfunction and thyroid carcinoma [16,17,26,27]. Therefore, it is important to understand the toxic effects of BDE209 on thyroid lesions and their underlying mechanisms.

We found that BDE209 has the potential to induce biphasic responses in normal human thyroid follicular cells (Nthy-ori 3-1), which indicates that the promotion of cell proliferation is manifested only when the exposure dose is under a certain threshold, while above this threshold, it inhibits cell growth. This result is consistent with several experimental studies showing apparent nonmonotonic effects of BDE209 in various cell lines and organisms [28,29,30,31]. In consideration of the fact that an environmentally relevant serum concentration of BDE209 in humans has a low threshold, we focused mainly on the effects and mechanism of low concentrations of BDE209 on the thyroid [19,32]. Moreover, long-term BDE209 exposure (≥27 days) was used to explore the chronic toxicity of BDE209 on normal human thyroid and the PTC cell lines, the effects of which are difficult to determine in experimental settings in humans [33].

Interestingly, we identified a biological relationship between BDE209 and the transcription factor TRß in the normal human thyroid follicular cell line, which indicates that there may be competitive inhibition between BDE209 and TRß, not only at the expressional level but also at the functional level. Here, we provide several lines of proof-of-principle evidence supporting our hypothesis. First, the chemical structure of BDE209 closely resembles T3 and T4 [34]. Hydroxylated metabolites of BDE209 present high affinity towards transport protein and bind to TRs [35,36]. BDE209 is the highest brominated PBDE, and its binding affinity with TRs increases significantly with bromination degree [36]. Second, TRs are ligand-dependent transcription factors mediating the actions of THs in development, growth, and differentiation. TRß is expressed predominantly in thyroid, kidneys, liver, brain and heart [37]. PBDEs congener could partially dissociate TR from the response elements acting through the DNA-binding domain [38]. Furthermore, analysis of multiple publicly accessible datasets showed an inverse association between the expression of TRß and PTC, which filled this gap. Therefore, we propose that BDE209 induces thyroid tumourigenesis via the inhibition of TRß at multiple levels.

The role of endocrine-disrupting chemicals in thyroid carcinogenesis has recently become increasingly considered [39]. Some researchers proposed that the thyroid-disrupting activity may promote thyroid carcinogenesis through a thyroid-stimulating hormone-mediated hyperproliferation of the thyrocytes [40]. Our study determined the direct toxic effects of BDE209 on TRß both in vitro and in vivo, which suggests that BDE209 may promote thyroid carcinogenesis through a TRß-mediated pathway. However, real-life environmental exposure is characterized by simultaneous biocontamination with multiple pollutants, which may interact through additive, synergistic, or antagonistic mechanisms [41]. This warrants further investigation.

## 5. Conclusions

In summary, long-term exposure to environmental concentrations of BDE209 could inhibit TRß expression and function, causing chronic toxicity and potential thyroid tumourigenesis. However, more research is necessary to estimate the toxicity of BDE209 in its administration and applications.

## Figures and Tables

**Figure 1 cancers-14-02772-f001:**
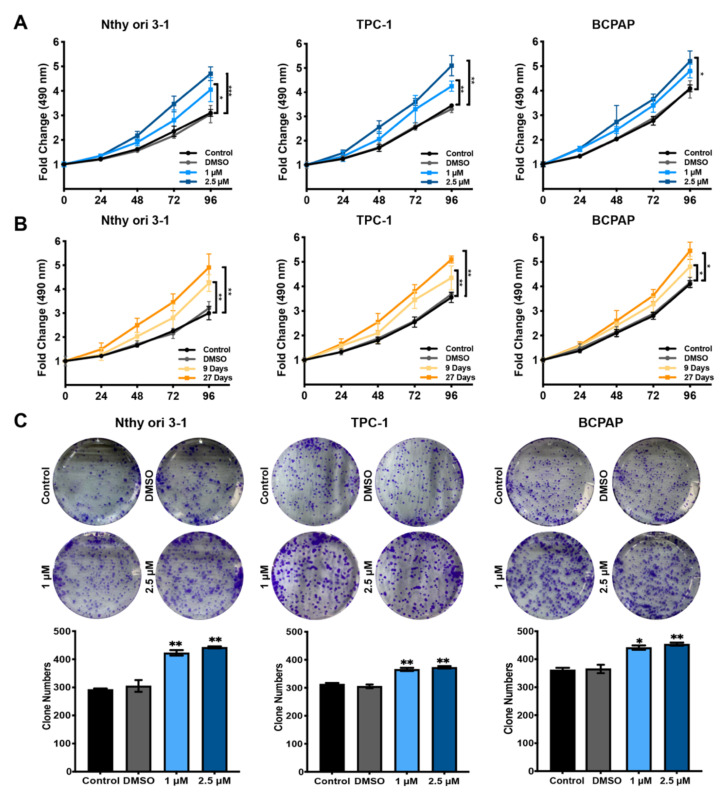
Hormesis effect of BDE209 on Nthy-ori 3-1 and PTC cell proliferation. (**A**) Cell viability in the indicated cell lines and concentration was assessed by MTS assays, and the results were calibrated to time 0. (**B**) Cell viability was measured in the indicated cell lines under decabromodiphenyl ether (BDE209) treatment by MTS assays. Nthy-ori 3-1, TPC-1, and BCPAP cells were treated with 2.5 μM BDE209 for 9 and 27 days. The DMSO group of each cell line was treated with DMSO at a concentration of 2.5 μM for 27 days. (**C**) Colony formation assay of the indicated cells treated with BDE209 or DMSO. * *p* < 0.05, ** *p* < 0.01, *** *p* < 0.001 based on Student’s *t*-test. The data shown are the means ± SD of triplicate wells and are representative of at least three replicate experiments.

**Figure 2 cancers-14-02772-f002:**
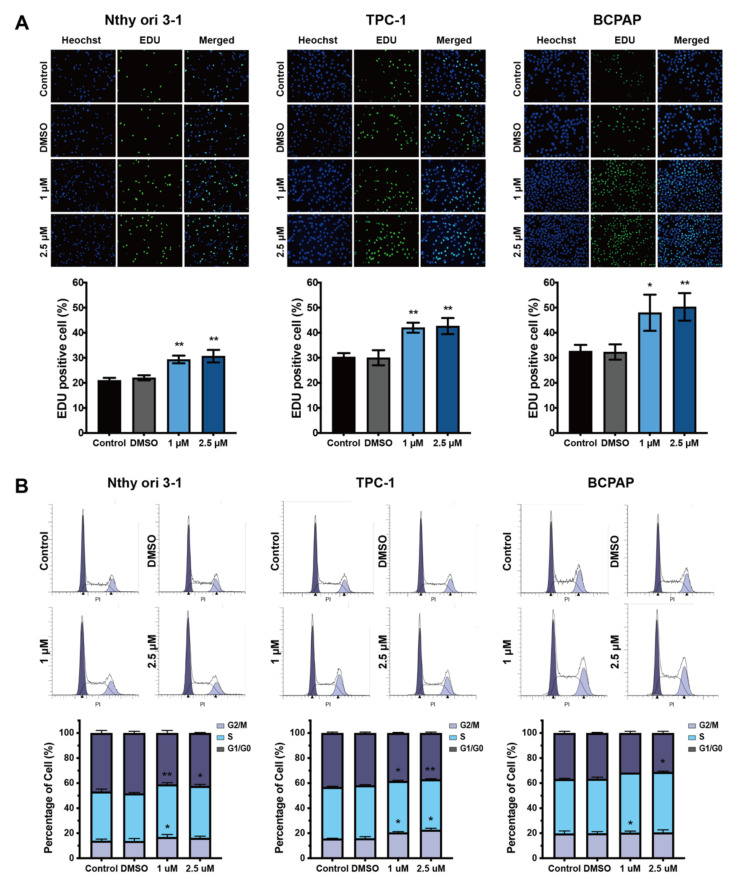
Cell cycle analysis of Nthy-ori 3-1 and PTC cells after exposure to BDE209 (1 and 2.5 μM). (**A**) Representative images (top panel) and quantitative results (bottom panel) of EdU assays of Nthy-ori 3-1, TPC-1 and BCPAP cells. (**B**) Percentages of the cells in the cycle phase of the flow cytometry analysis after treatment with BDE209 or DMSO. * *p* < 0.05; ** *p* < 0.01. Each experiment was repeated at least three times.

**Figure 3 cancers-14-02772-f003:**
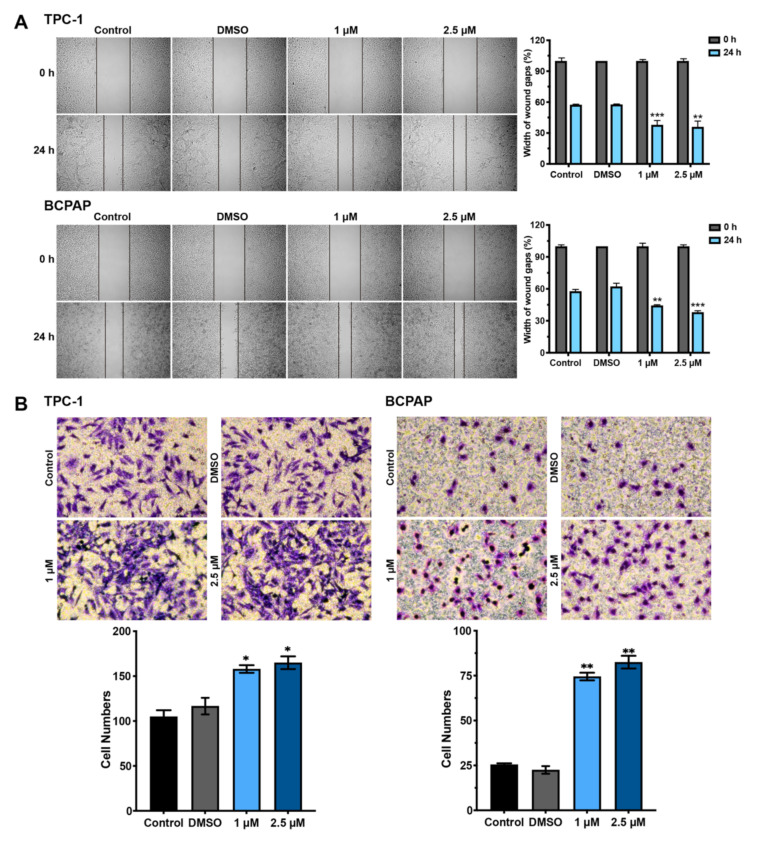
Hormesis effect of BDE209 on Nthy-ori 3-1 and PTC cell invasion and migration. (**A**) Wound-healing assay was performed to assess the mobility of TPC-1 and BCPAP cells treated with different concentrations of BDE209 compared with the negative controls. Representative images are shown on the left, and the statistical analysis is shown on the right. (**B**) The effects of RUVBL1 on cell migration. TPC-1 and BCPAP cell lines were treated as indicated and evaluated by Transwell migration assays. Top panel: Representative images of cell migration. Bottom panel: Quantitative results of the migration assays. * *p* < 0.05, ** *p* < 0.01, and *** *p* < 0.001, based on Student’s t-test. Each experiment was repeated at least three times.

**Figure 4 cancers-14-02772-f004:**
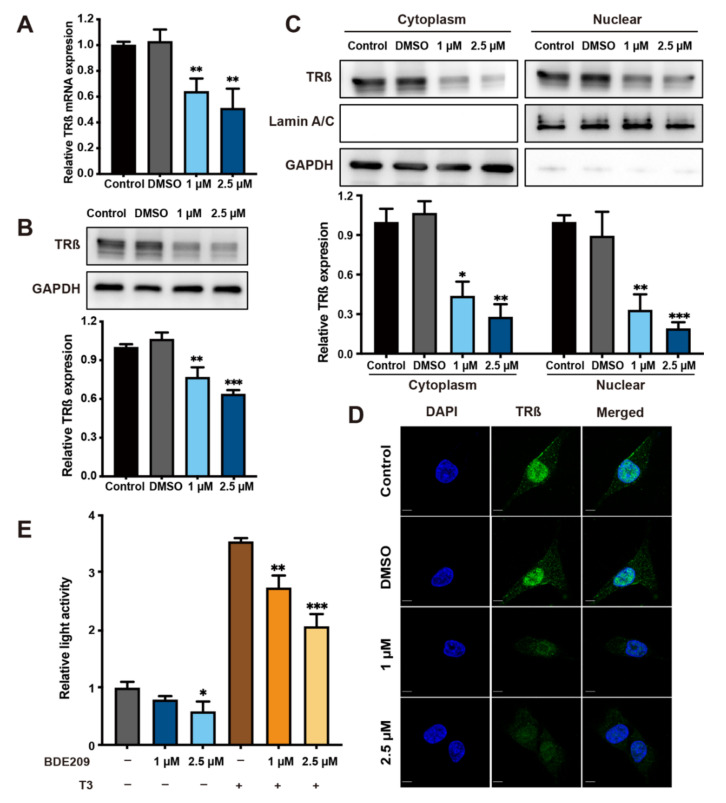
BDE209 promotes tumourigenesis by inhibiting TRß expression and function. (**A**) Nthy-ori 3-1 cells were treated with 1 and 2.5 μM BDE209 for 27 days and compared with negative controls. TRß mRNA levels were detected at the indicated time points by qRT–PCR. (**B**) Nthy-ori 3-1 cells were treated with 1 and 2.5 μM BDE209 for 27 days. After the cells were harvested, the whole lysates were subjected to Western blotting. Upper panel: Representative images of Western blotting. Bottom panel: Quantitative results of Western blotting from triplicate experiments. (**C**) Cytoplasmic and nuclear TRß protein levels were analysed in Nthy-ori 3-1 cells. GAPDH and Lamin A/C were used as cytoplasmic and nuclear protein-loading controls, respectively. Quantitative results of Western blotting from triplicate experiments are shown at the bottom. (**D**) TRß (green) localization after treatment with different doses of BDE209 was determined using immunofluorescence staining. All pictures were imaged using a confocal microscope. Representative images are shown with a 10 μm scale bar. (**E**) The plasmids pTRE-Luc and pRL-TK were cotransfected into 293T cells. The cells were incubated with or without 0.1 μM T3 and BDE209 for 48 h. Data are the mean ± SD of experiments performed in triplicate. * *p* < 0.05, ** *p* < 0.01, and *** *p* < 0.001. Each experiment was repeated at least three times.

**Figure 5 cancers-14-02772-f005:**
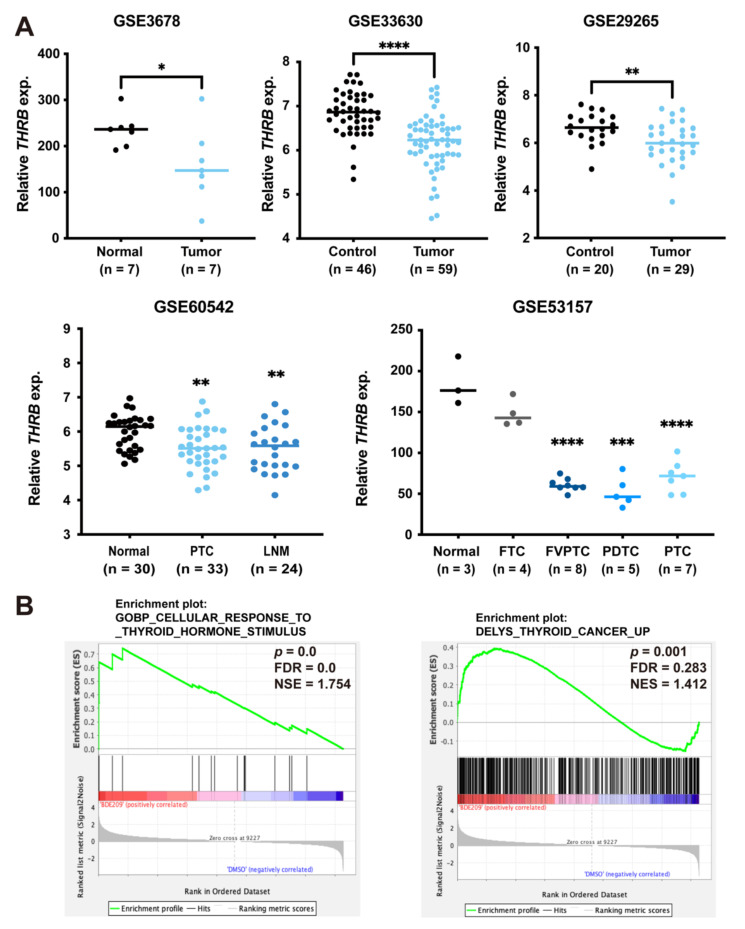
Inverse association between TRß expression and thyroid cancer. (**A**) Relationship between *THRB* and clinical outcomes. Thyroid tumours in GSE3678, GSE33630, GSE29265 and GSE60542 were analysed. * *p* < 0.05, ** *p* < 0.01, *** *p* < 0.001 and **** *p* < 0.0001. PTC, papillary thyroid carcinoma; LNM, lymph node metastasis; FTC, follicular thyroid carcinoma; FVPTC, follicular variant papillary thyroid carcinoma; PDTC, poorly differentiated thyroid carcinoma. (**B**) The enrichment of the indicated gene sets by GSEA.

**Figure 6 cancers-14-02772-f006:**
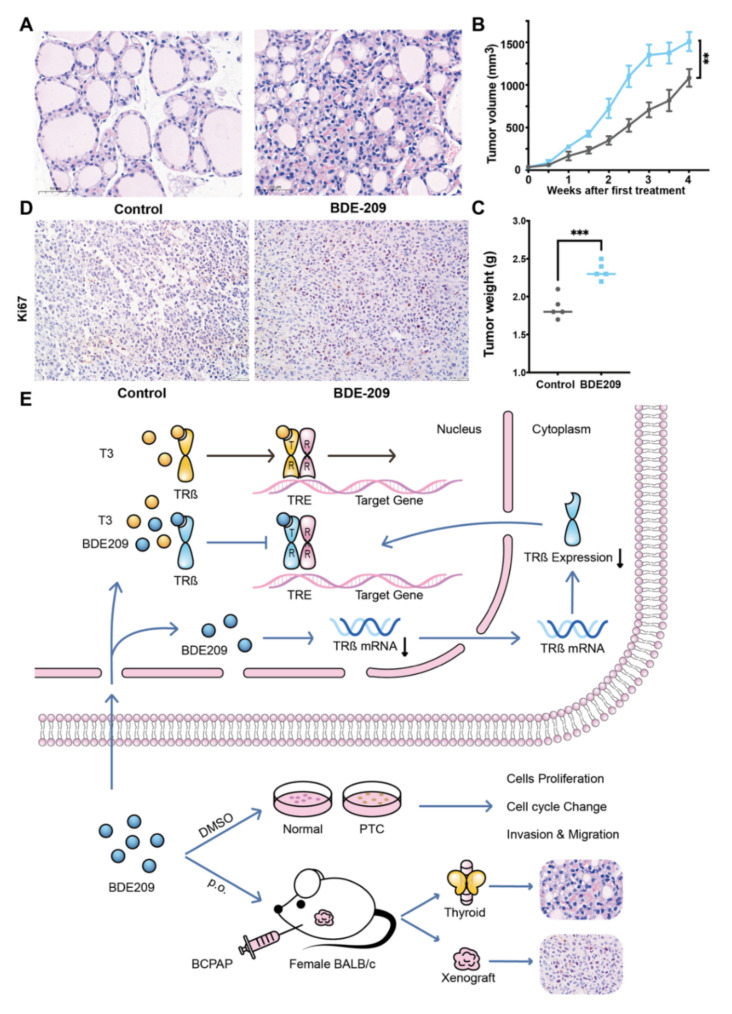
BDE209 promotes the proliferation of thyroid follicular cells and BCPAP cells in BALB/c female mouse xenograft models. (**A**) Mice bearing BCPAP xenografts were treated with control (peanut oil p.o.) or high dose BDE209 (2000 µg/(kg/d) p.o.) for 4 weeks (*n* = 5/group). Proliferative thyroid follicles with an increased number of follicular cells were observed in the BDE209 group compared with the control group. Scale bars, 50 μm. (**B**) Tumour volumes with or without BDE209 exposure were measured twice per week. (**C**) Tumours were collected and weighed when the mice were sacrificed. (**D**) Immunohistochemical staining of the tumours with or without BDE209 exposure is shown. Scale bars, 50 μm. (**E**) Working model of long-term exposure to decabromodiphenyl ether promotes the proliferation and tumourigenesis of papillary thyroid carcinoma by inhibiting TRß. ** *p* < 0.01, *** *p* < 0.001, based on Student’s t-test. Each experiment was repeated at least three times.

## Data Availability

The data presented in this study are available on request from the corresponding author.

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
