# Peer review of "Long-Term Exposure to Decabromodiphenyl Ether Promotes the Proliferation and Tumourigenesis of Papillary Thyroid Carcinoma by Inhibiting TRß"

_cancers, 2022, doi:10.3390/cancers14112772_

Round 1

Reviewer 1 Report

Reviewer comments:

Comments to the Author

In this study, authors have investigated toxic effects of long-term exposure to environmental concentrations of BDE209 on the pathogenesis and progression of PTC because the effects of long-term exposure to BDE209 on the thyroid and the underlying mechanisms are unclear. This study determined that long-term exposure to BDE209 could cause chronic toxicity and potential tumorigenesis by inhibiting TRß, which induces the proliferation of thyroid tissue and thyroid carcinoma. These findings emphasize the damaging effects of BDE209 exposure on human thyroid and papillary thyroid carcinoma.

This manuscript is for the most part well written and discussed the recent studies. However, to increase the readability, authors are advised to incorporate some suggestions.

Minor criticisms

  • Direct introduction to BDE209 is misleading in the “Simple summary” section. Rewrite this section with introducing this compound.
  • Please undergo a thorough check of the manuscript for typographical and grammatical errors.

Reviewer 2 Report

Dear authors,

The paper Long-Term Exposure to Decabromodiphenyl ether Promotes 2 the Proliferation and Tumorigenesis of Papillary Thyroid 3 Carcinoma by Inhibiting TRß, containing an important and significant amount of research, molecular and genetic testing of the impact of PBDEs and endocrine disruptors of tumorigenesis is worth the publication. The impressive experimental research bring enough solid contribution to the scientific community. 

Good luck! 
